# Inference of recombination maps from a single pair of genomes and its application to ancient samples

**Gustavo V. Barroso**[ID]*, **Nataša Puzović**[ID], **Julien Y. Dutheil**[ID]

Max Planck Institute for Evolutionary Biology, Department of Evolutionary Genetics, August-Thienemann-Straße , Plön–GERMANY

* gvbarroso@evolbio.mpg.de

**Data Availability Statement:** Scripts necessary to reproduce the analyses in this study are available as S1 File. The recombination maps for ancient hominins generated in this work are made available as BedGraph files, which can be visualized in the

## Abstract

Understanding the causes and consequences of recombination landscape evolution is a fundamental goal in genetics that requires recombination maps from across the tree of life. Such maps can be obtained from population genomic datasets, but require large sample sizes. Alternative methods are therefore necessary to research organisms where such datasets cannot be generated easily, such as non-model or ancient species. Here we extend the sequentially Markovian coalescent model to jointly infer demography and the spatial variation in recombination rate. Using extensive simulations and sequence data from humans, fruit-flies and a fungal pathogen, we demonstrate that iSMC accurately infers recombination maps under a wide range of scenarios–remarkably, even from a single pair of unphased genomes. We exploit this possibility and reconstruct the recombination maps of ancient hominins. We report that the ancient and modern maps are correlated in a manner that reflects the established phylogeny of Neanderthals, Denisovans, and modern human populations.

## Author summary

In sexually-reproducing species, meiotic recombination causes the genome of each individual to be a mosaic of DNA sequences that existed in its ancestral population. As a result, genealogical ancestry changes along the genome and shapes genetic diversity. The importance of recombination in genome evolution has motivated a surge in the development of statistical tools to infer genome-wide variation in the recombination rate using polymorphism data. For the most part, however, these methods rely on relatively large sample sizes. Here, we introduce iSMC–a new tool that infers recombination maps while simultaneously modelling the demographic history. A critical improvement over existing methods is that iSMC has high accuracy using as little as a single diploid genome. Using experimentally derived recombination maps from fruit-flies and a fungal pathogen, we demonstrate that iSMC compares well to state-of-the-art methods that require larger sample sizes. We further analyse data from ancient hominins, showcasing that our method can extract information in intrinsically limited datasets. These results suggest that iSMC is

UCSC genome browser. They have been deposited as a public FigShare project with the following DOI: Ust'Ishim: 10.6084/m9.figshare.7182776. Vindija Neandertal: 10.6084/m9.figshare.7182809 Altai Neandertal: 10.6084/m9.figshare.7182785. Denisovan: 10.6084/m9.figshare.7182794. All hominin together: 10.6084/m9.figshare.8269703. Time-restricted recombination maps: 10.6084/m9.figshare.9943856. The iSMC software can be accessed at https://github.com/gvbarroso/iSMC.

**Funding:** JYD acknowledges funding from the Max Planck Society. This work was supported by a grant from the German Research Foundation (Deutsche Forschungsgemeinschaft) attributed to JYD, within the priority program (SPP) 1590 "probabilistic structures in evolution". The funders had no role in study design, data collection and analysis, decision to publish, or preparation of the manuscript.

**Competing interests:** The authors have declared that no competing interests exist.

a valuable tool that can foster studies in non-model organisms. Moreover, its joint-inference approach of demography and the recombination landscape represents a step towards more realistic models in population genomics.

## Introduction

Meiotic recombination is a major driver of the evolution of sexually-reproducing species [1]. The crossing-over of homologous chromosomes creates new haplotypes and breaks down linkage between neighbouring loci, thereby impacting natural selection [2,3] and consequently the genome-wide distribution of diversity [4]. The distribution of such cross-over events is heterogeneous within and among chromosomes [5,6], and commonly referred to as the recombination landscape–a picture of how often genetic variation is shuffled in different parts of the genome. Interestingly, this picture is not static, but instead is an evolving trait that varies between populations [7,8] and species [9]. Moreover, the proximate mechanisms responsible for shaping the recombination landscape vary among *taxa*. For example, among primates (where the *PRDM9* gene is a key player determining the location of so-called recombination hotspots [10]) the landscape is conserved at the mega-base (Mb) scale, but not at the kilo-base (kb) scale [11]. In birds, which lack *PRDM9*, the hotspots are found near transcription start sites in the species that have been studied so far [8,12]. In *Drosophila* (where clear hotspots appear to be absent [13]), inter-specific changes are associated with mei-218 variants [14], a gene involved in the positioning of double-strand breaks [15]. The molecular machinery influencing the distribution of cross-over events is still poorly understood in many other groups, where a survey of the recombination landscape in closely related species is lacking.

Aside from their intrinsic value in genetics, accurate recombination maps are needed to interpret the distribution of diversity along the genome. Since the rate of recombination determines the extent to which linked loci share a common evolutionary history [16], inferring selection [17–19], introgression [18,20] and identifying causal loci in association studies requires knowledge of the degree of linkage between sites [21]. Furthermore, recombination can cause GC-biased gene conversion [22,23], which can mimic the effect of selection [24] or interfere with it [25]. Although important, obtaining recombination maps remains a challenging task. Due to the typically low density of markers, experimental approaches provide broad-scale estimates and are limited in the number of amenable *taxa*. Conversely, population genomic approaches based on coalescent theory [26,27] have proved instrumental in inferring recombination rates from polymorphism data.

Traditionally, population genomic methods infer recombination maps from variation in linkage disequilibrium (LD) between pairs of single nucleotide polymorphisms (SNPs) [28–30]. However, since "LD-based" methods typically require large sample sizes per population (from a dozen haplotypes [31]), their application is restricted to a few model organisms where such data are available. Here we introduce a new modeling framework (iSMC) to infer the variation in the recombination rate along the genome, using a single pair of unphased genomes. Using simulations, we show that iSMC is able to accurately recover the recombination landscape under diverse scenarios. We further demonstrate its efficacy with case studies in Humans, Fruit-flies and the fungal pathogen *Zymoseptoria tritici*, where experimental genetic maps are available. Finally, we exploit our new method to investigate the recombination landscape of ancient hominin samples: Ust'Ishim, the Vindija Neanderthal, the Altai Neanderthal, and the Denisovan. Because it allows inference from datasets for which sample size is intrinsically limited, such as ancient DNA samples, our method opens a new window in the study of the recombination landscape evolution.

## Results

### Overview of iSMC

Besides its common interpretation as a backwards-in-time process, the coalescent with recombination [32,33] can also be modelled as unfolding spatially along chromosomes [34]. Starting from a genealogy at the first position of the alignment, the process moves along the chromosome, adding recombination and coalescence events to the ensuing ancestral recombination graph (ARG) [35,36] (**Fig 1A**). Due to long-range correlations imposed by rare recombination events that happen outside the ancestry of the sample (in so-called trapped non-ancestral material [37]), the genealogy after a recombination event cannot be entirely deduced from the genealogy before, rendering the process non-Markovian. The sequentially Markovian coalescent process (SMC) [38,39] ignores such recombination events but captures most of the properties of the original coalescent [40] while being computationally tractable. This model is the foundation of recent tools for demographic inference [41–43] and has been used to infer the broad-scale recombination map of the human-chimpanzee ancestor based on patterns of incomplete lineage sorting [44,45].

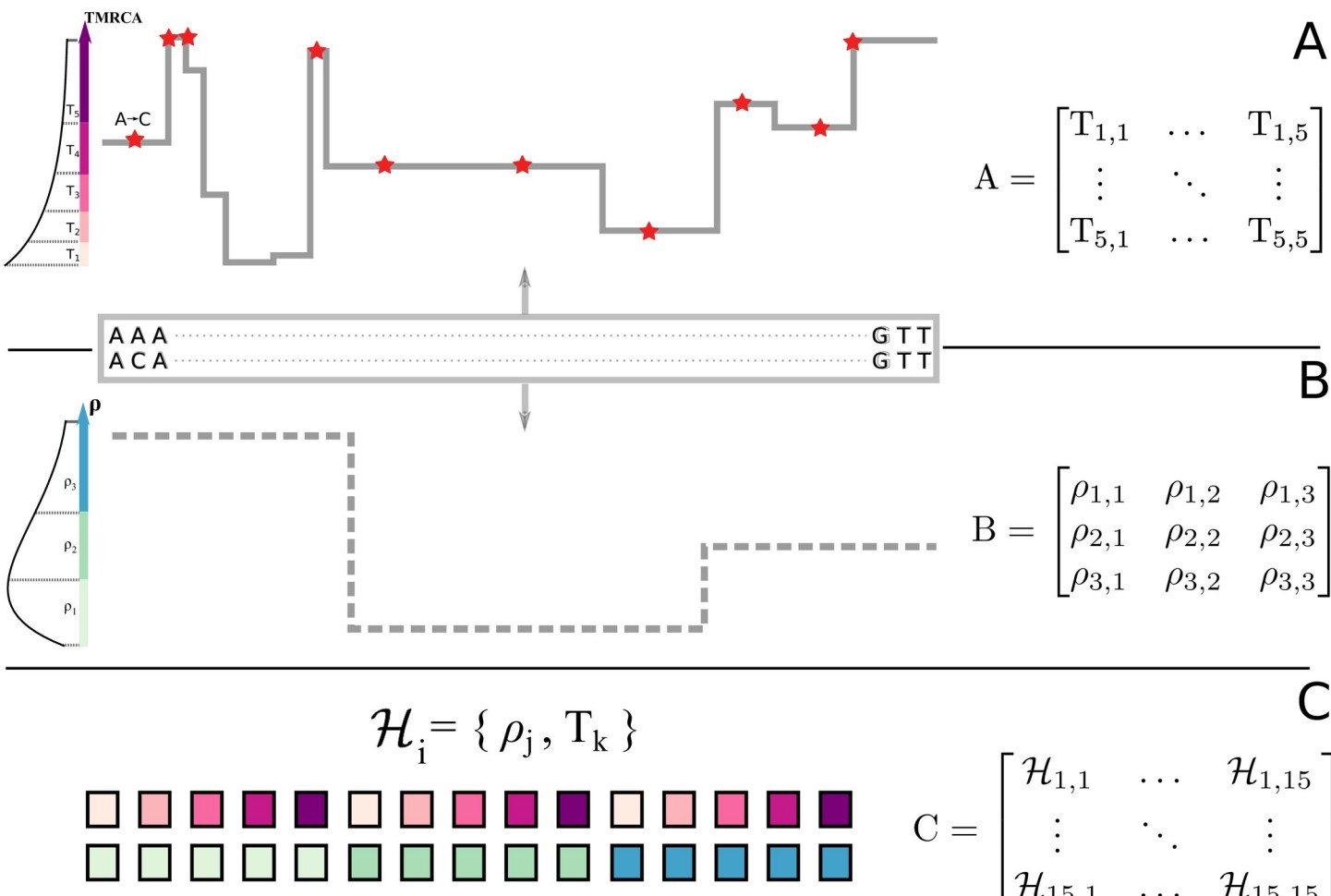

**Fig 1. Schematic representation of iSMC for one pair of genomes, with five time intervals and three recombination rate categories. A,** In the SMC process, the spatial distribution of TMRCAs can be described by a matrix of transition probabilities that depend on the population recombination rate ρ and the ancestral coalescence rates. **B,** variation in ρ along the genome, modelled as a Markovian process and described by a matrix of transition probabilities. **C,** the combination of both Markovian processes leads to a Markov-modulated Markovian process. The hidden states of the resulting hidden Markov model are all pairwise combinations of discretized classes in **A** and **B**.

In the SMC, transition probabilities between genealogies are functions of ancestral coalescence rates and–of key relevance to this study–the population recombination rate (ρ) [42,43]. An important limitation of current implementations, however, is that they either neglect variation in ρ along the genome and model its genome-wide average or they require the recombination landscape to be input in order to aid inference of other population genetic parameters (e.g., [46]). Heterogeneous recombination landscapes affect the SMC process by modulating the frequency of genealogy transitions: genomic regions with higher recombination rate are expected to harbour relatively more genealogies than regions with lower recombination rate (**Fig 1**). We leverage this information by extending the SMC to accommodate spatial heterogeneity in ρ (see Methods). In brief, our new model combines the discretised distribution of times to the most recent common ancestor (TMRCAs) of the pairwise SMC [42] (**Fig 1A**) with a discretised distribution of ρ (**Fig 1B**) to jointly model their variation along the genome. We model the transition between discretised ρ categories as another spatially Markovian process along the genome. Therefore, combining the SMC with the Markov model of recombination variation leads to a Markov-modulated Markov model. We cast it as a hidden Markov model [47,48] (HMM) to generate a likelihood function, where the observed states are orthologous nucleotides and the hidden states are {TMRCA, ρ-category} pairs (**Fig 1C**). We name our new approach "integrative sequentially Markov coalescent (iSMC)", as it enables jointly capturing the effect of time and space in the coalescent. Importantly, in contrast with methods that can take a pre-specified demographic history into account when reconstructing the recombination map [49,50], iSMC simultaneously infers coalescence rates and spatial variation in the recombination rate, potentially leading to more accurate estimates. This framework explicitly connects the genealogical process with the classical definition of LD as the non-random association of alleles at different loci [51], which has been formulated in terms of covariances in coalescence times [52]. Henceforth, we restrict the use of the term LD to its "topological" interpretation [53].

iSMC is based on a neutral model where time is re-scaled and measured in units of the effective population size (Ne). Thus, information about the recombination rate is obtained in the form of the compound parameter $\rho = 4 \times Ne \times r$. Since under neutrality and panmixia, Ne is constant along autosomes, we use the inferred ρ landscape as a proxy for the spatial variation in the molecular rate *r* that measures the historical rate of cross-over events per-nucleotide per generation. (Note that local variation in TMRCA primarily reflects genealogical and sampling variance and cannot, on its own, be used to tease apart Ne and *r*.) Our approach is to model spatial variation in ρ using a Gamma distribution with *k* discretised categories (see Methods). Setting *k* equal to 1 is equivalent to the "standard" PSMC' where ρ is homogeneous along the genome [43], and can be used as a null model to formally test the existence of spatial variation in the recombination rate. To this end, after fitting both homogeneous and heterogeneous models to the data, Akaike's Information Criterium (AIC) [54] is employed as a means of model selection. If AIC favours a spatially heterogeneous model over the null model where ρ is constant along the genome, iSMC can then be used to estimate a recombination landscape of single-nucleotide resolution by weighting the discretised values of the Gamma distribution of ρ with their local posterior probabilities. In the following section, we benchmark our model on different simulated scenarios. Therein, we computed the proportion of variance in simulated maps that is explained by inferred maps ($R^2$) after binning the landscapes into windows of 50 kb, 200 kb, 500 kb and 1 Mb by averaging single-nucleotide estimates of ρ.

## Simulation study

To assess iSMC's overall performance, we simulated five recombination landscapes corresponding to different patterns of magnitude and frequency of change in ρ and a "null" scenario

with constant recombination rate along the genome (see Methods). For each of the five scenarios, we simulated 10 replicate ARGs using SCRM [55], each describing the ancestry of 2 haploid chromosomes. When fitting a model to these simulated data, we tested two discretisation schemes for the joint distribution of TMRCAs and Gamma-distributed recombination rates (see Methods): the first with 40 time intervals, five ρ categories; the second with 20 time intervals, 10 ρ categories, leading to a total of 200 hidden states in both configurations. Model selection based on AIC favours the correct model in 45 of the 50 datasets (**S1 Table**), with the five exceptions belonging to the scenario where changes are frequent and of small magnitude. In this regime, transitions to regions of slightly different recombination rates do not significantly skew the distribution of genealogies, and the short length of blocks with constant ρ leaves little signal in the data. Accordingly, among the four heterogeneous landscapes that we considered, median $R^2$ is 38%, 44%, 49% and 51% for increasing window sizes in the five identifiable replicates of such challenging scenario, and 66%, 77%, 77% and 83% for increasing window sizes in the other three (**Fig 2A**, **S2 Table**). Overall, the results are consistent between replicates and robust to the choice of discretisation, although the 40x5 configuration performs better in the scenario with the challenging parameter combination (**Fig 2A**). Therefore, in the following we focus on the 40x5 configuration, noting that it implements a finer discretisation of time that is more adequate to capture the effect of ancestral demography. As we introduce new simulated scenarios, we focus on the recombination landscape with frequent changes of large magnitude.

## Demographic history

The random sampling of haplotypes during population bottlenecks and expansions affects LD between SNPs, thus creating spurious signals of variation in ρ [49,56,57]. To test whether iSMC could capture the effect of demography on the inference of recombination maps, we simulated a heterogeneous recombination landscape coupled with either a recent 20-fold increase or ancient 20-fold decrease in population size (see Methods). We then fitted our model twice for each scenario: first, erroneously assuming a flat demographic history; second, allowing iSMC to infer piece-wise constant coalescence rates in order to accommodate population size changes. Overall, $R^2$ is high (median 67%, 78%, 79% and 83% for increasing window sizes in all scenarios, **Fig 2B and 2C**, **Fig 3**), showing that the inferred recombination landscape is relatively robust to misspecification of the demographic scenario, but is systematically higher when demography is jointly inferred (55%, 69%, 75% and 79% when demography is assumed constant versus 71%, 79%, 80% and 85% when demography is jointly inferred, **Fig 2B and 2C**, **S3 Table**). The difference is stronger at the fine scale, where, in the presence of complex demography the distribution of genealogies can get locally confined to a time period, and ignorance about differential coalescence rates reflects poorly on local ρ estimates. We conclude that the joint-inference approach of iSMC can disentangle the signal that variable recombination and fluctuating population sizes leave on the distribution of SNPs.

## Introgression events

Recent studies suggested that introgression is a frequent phenomenon in nature [58,59]. The influx of a subset of chromosomes from a "source" into a "target" population (in a process analogous to a genetic bottleneck) introduces long stretches of SNPs in strong LD. Past introgression events will thus affect runs of homozygosity, biasing the distribution of genealogies. To test the robustness of iSMC to the confounding effect of introgression, we simulated two scenarios of admixture which differ in their time of secondary contact between populations (see Methods). The proportion of variance explained remains high (median 57%, 64%, 69% and 75% for increasing window sizes, **Fig 2D**, **S4 Table**) and depends on the time when

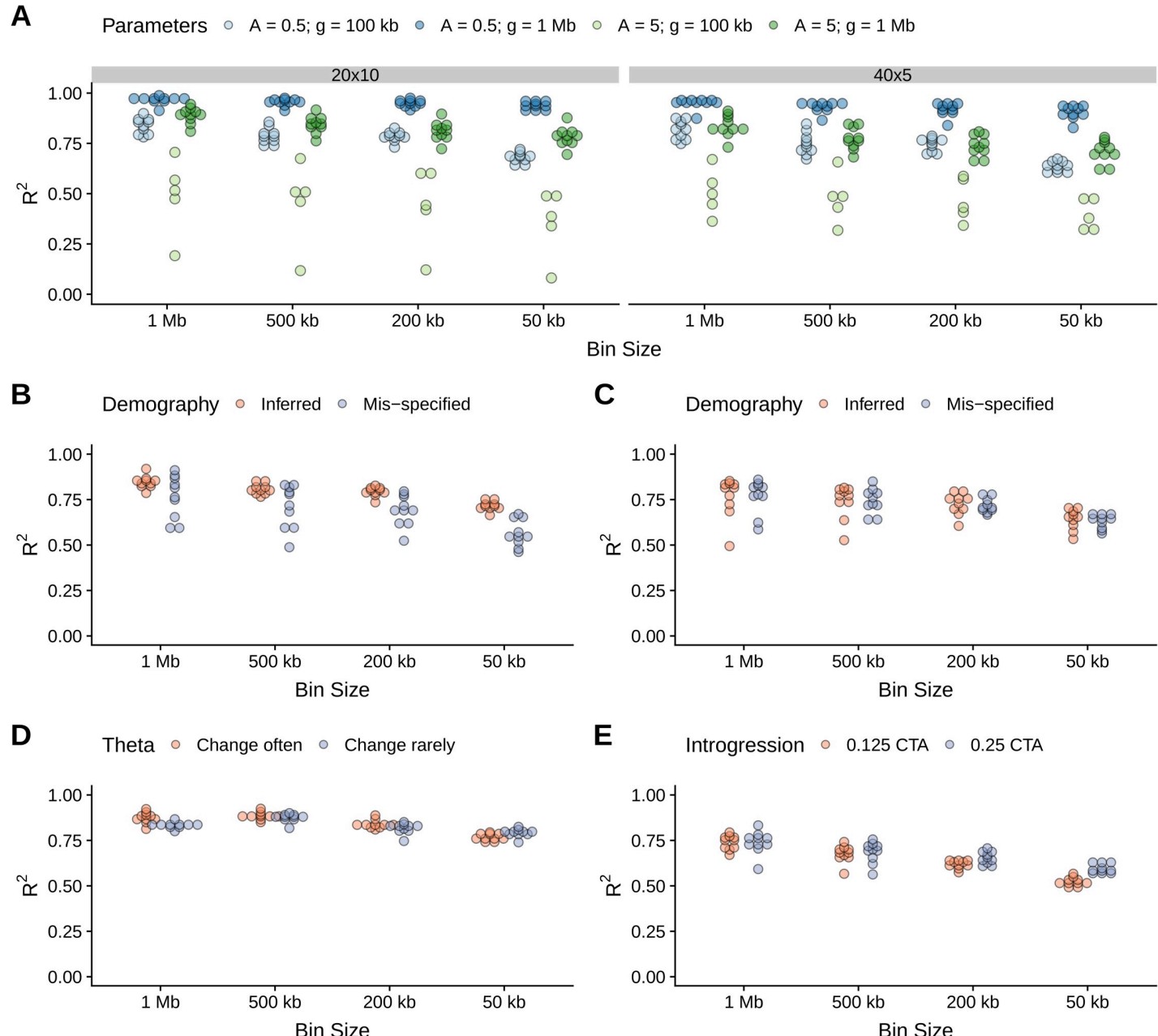

**Fig 2. Recombination map recovery under various simulated scenarios according to bin size.** Dot plots show the distribution of squared Pearson correlation coefficients ($R^2$) between the simulated and inferred recombination maps. See Methods section for details about the simulations. **A**, four scenarios of spatial variation in the recombination rate, corresponding to different combinations of parameters (colour, see text), and comparison between two discretisation schemes (panels). **B-C**, comparison between a model where demography is mis-specified and another where it is jointly inferred (colour), in scenarios of recent growth (**B**) or ancient bottleneck (**C**). **D**, two scenarios of spatial variation in the mutation rate, varying its frequency of change (colour). **E**, two scenarios of introgression, varying the time of gene-flow (colour, CTA stands for Coalescent Time units Ago).

introgression occurred. Recombination maps are less accurately recovered in case of recent introgression (52%, 62%, 68% and 75% versus 58%, 65%, 70% and 75% for increasing window sizes), because in such case there has been less time for recombination to break SNP associations that do not reflect local ρ in the target, sampled population.

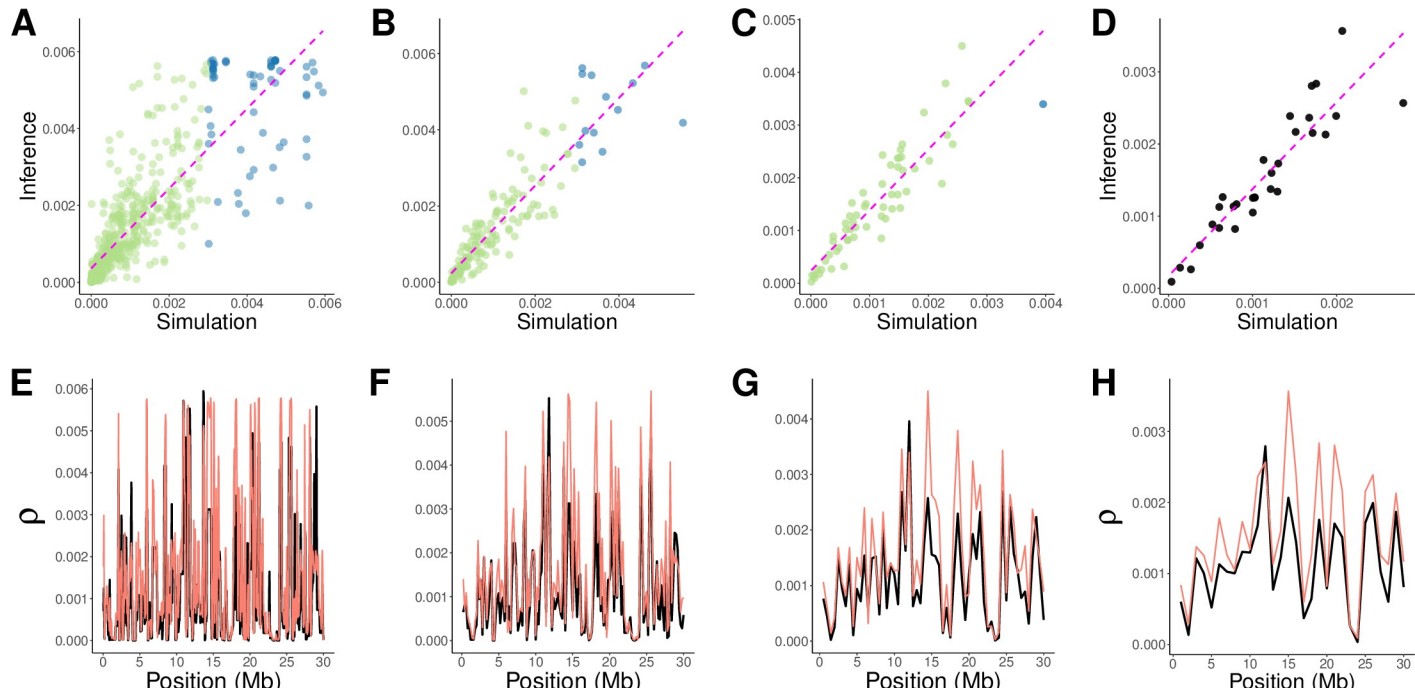

**Fig 3. Inference of recombination maps in the presence of recent population growth.** Each column represents a different bin size (increasing from left to right, 50 kb, 200 kb, 500 kb and 1 Mb). **A-D**, scatter-plots of inferred versus simulated maps, coloured according to the θ / ρ ratio (green > = 1, blue < 1). The dashed magenta line represents ordinary least squares regression. **E-H**, corresponding simulated (black) and inferred (orange) maps.

## Variation in mutation rate

The rate of *de novo* mutations varies along the genome of many species. For example, CpG dinucleotides experience an increase in mutation rate (μ) as a result of methylation followed by deamination into thymine, whereas the efficiency of the molecular repair machinery is negatively correlated with the distance from the DNA replication origin, causing μ to vary accordingly [60]. Such heterogeneity could bias iSMC's estimates because the transition into a region of higher μ mimics the transition to a genealogy with a more ancient common ancestor since in both cases the outcome is locally increased genetic diversity. To assess the impact of variation of mutation rate on the estimation of recombination rate, we simulated two scenarios of variation of $\theta = 4 \times Ne \times \mu$ along the genome, corresponding to low and high frequencies of change, relative to the frequency of change in the recombination rate. We report that transitions to different mutation rates along the genome globally do not introduce substantial biases in our estimates (median $R^2$ 76%, 83%. 88% and 88% for increasing window sizes when the mutation landscape changes often, and 80%, 83%, 88% and 83% for increasing window sizes when the mutation landscape changes rarely, **Fig 2E**, **S5 Table**).

We also investigated the impact of the genome-wide average mutation rate μ on iSMC's accuracy. Since SNPs are informative about the distribution of genealogies along the genome, the ability of population genetic methods to identify ancestral events depends on their abundance. In this case, the actual parameter of interest is the θ / ρ ratio, where values smaller than 1 imply less mutation than recombination events in the history of the sample (meaning some cross-overs will be effectively "invisible" in the sequence data) and values higher than 1 imply more mutations per recombination, thereby increasing resolution in our inference. As expected, we find that accuracy is reduced for a θ / ρ ratio equal to 0.5 (median $R^2$ 40%, 47%, 50% and 60% for increasing window sizes, **S2 Fig**). Reassuringly, however, iSMC retains high

accuracy for ratios > = 1.0, (median $R^2$ 51%, 56%, 61% and 63%; 60%, 74%, 75% and 75%; 75%, 79%, 82% and 83% for increasing ratios and window sizes, **S2 Fig**), especially starting from 1.5 –a value commonly assumed for human data [61].

### Comparison with state-of-the-art methods

Simulation studies are useful to understand the theoretical limitations of a method under ide-alised conditions. On the other hand, inference in real datasets provides the most realistic assessment of performance. For this reason, we benchmarked iSMC in two organisms with contrasting genomic architectures and evolutionary histories, and for which experimental genetic maps are available [62,63]. Using a 40x5 configuration of hidden states, we fitted iSMC independently to each of three pairs of genomes from each species (see Methods). In all six samples, AIC favoured a heterogeneous recombination landscape. We then computed $R^2$ between the iSMC-inferred maps and the experimental genetic maps and used these as a proxy for iSMC's accuracy. In the *Z. tritici* samples $R^2$ estimated with iSMC maps are **24%** (+/- 13%), **30%** (+/- 12%) and **38%** (+/- 16%) respectively at the 20 kb scale for chromosome 1. Using the same chromosome and window size, the recombination map obtained with LDhat by Stuken-brock and Dutheil [64] with 13 haploid sequences (of which our samples are a subset) explains **25.35%** of the variance in the same genetic map. In the *D. melanogaster* samples, $R^2$ estimated with iSMC maps are **71%** (+/- 27%), **56%** (+/- 19%) and **73%** (+/- 22%) respectively at the 1 Mb scale for chromosome 2L. Using the same chromosome and window size, the recombina-tion maps obtained by Chan et al. [30] with LDhelmet and by Adrion et al. [50] with ReLERN with 10 haploid sequences explain **55%** and **46%** of the variance in the same genetic map, respectively. We thus conclude that remarkably, iSMC can outperform methods that use larger samples sizes in both *Z. tritici* and *D. melanogaster*, which are species with a large density of SNPs.

### Application to modern and ancient human samples

The positioning of double-strand breaks leading to cross-over events is directed by several pro-teins that exert their influence at different scales. Evolution of such proteins results in differen-tiation of recombination maps with an increasing divergence between populations and species [65]. To gain insight into the evolution of the recombination landscape in hominins, we built a dataset of 22 individuals from 11 modern human populations sampled in the Simons Genome Diversity Panel [66] together with four ancient samples [67]: the Altai Neanderthal [68], the Vindija Neanderthal [69], the Denisovan [70] and the Ust'Ishim individual [61], a 45,000-year-old modern human from Siberia. Due to the overall low polymorphism in hominins, we sought to increase accuracy in our inference by fitting iSMC to whole-genome sequences of each individual using a 40x10 hidden states configuration. Estimation of the parameters of the prior distributions illustrated in **Fig 1B** (the genome-wide average recombination rate ρ, the shape of the Gamma distribution α and the transition probability between recombination cate-gories δ, see Methods) yielded consistent values across extant samples (**S1 Fig**). Conversely, jointly inferring such parameters in the ancient samples proved challenging because of wide-spread missing data. To address this issue, we first inferred the demographic parameters in each of the four ancient samples with a homogeneous recombination model (40x1 hidden states configuration). Then, we obtained the recombination maps of the ancient samples using the resulting sample-specific coalescence rates together with extant-human estimates of ρ, α and δ. Although this procedure implies that extant and ancient samples share their prior distri-butions of recombination rate variation, computing posterior probabilities with an HMM con-fronts such distributions with the data such that the resulting ancient maps should reflect their

specific characteristics of LD. We note that any biases introduced by this procedure should increase the similarity between modern and ancient recombination maps. The reconstructed phylogenetic distance between maps of modern and ancient hominins outlined below is therefore conservative.

To quantify the similarity among these recombination landscapes, we computed Spearman's rank correlation ($r_s$) between all pairs of individual maps at the 50 kb, 200 kb and 1 Mb scales. We further added the sex-averaged genetic map from DECODE [7] to the dataset in order to assess iSMC's accuracy (**Fig 4A, 4C and 4E**). Notwithstanding the low density of SNPs along single human genomes, correlations between the maps from iSMC and DECODE increase rapidly with window size. We then used $1 - r_s$ as a measure of the pair-wise distance between maps to construct dendrograms (**Fig 4B, 4D and 4F**) and discovered that the evolution of the recombination landscape globally reflects the currently accepted evolutionary history of hominins. Specifically, at the 50 kb and 200 kb scales, the major geographic groups are recovered with high bootstrap support (with the exception of the Punjabi individuals, see argumentation below): Africans, Asians and Europeans each form a distinct cluster; the 45,000-year-old Ust'Ishim is sister to modern humans, depicting ancestral similarities that have been frozen by his demise soon after the out-of-Africa migration; and all modern humans have diverged from the monophyletic Neanderthal-Denisovan group (**Fig 4B and 4D**). We note that, while Ust'Ishim is branching together with other *H. sapiens* samples in a well supported group, it is not placed immediately outside the Europe + Asian clade, as expected given that he is part of the out-of-Africa migration. One possibility is that the high level of missing data from this sample increases the noise in the inferred recombination map–therefore decreasing the similarity with other *H. sapiens* maps. Further analyses are necessary to pinpoint the precise reason for this branching pattern. Interestingly, the Punjabi individuals, who speak an Indo-European language, fall within the European clade. The Punjabi population contains a large proportion of Ancestral North Indian ancestry, a ghost population that shows similarity to West-Eurasian populations [66,71]. Moreover, African samples can be further differentiated at the population level with high confidence, likely because population structure (hence recombination landscape evolution and differentiation) extends deeper into the history of this continent compared to the non-African populations. At the 1 Mb scale, the differentiation among clades of modern human populations becomes blurrier (**Fig 4E and 4F**), as expected due to the slower evolution of the recombination landscape at larger scales. These results cannot be accounted for by patterns of LD left by population-specific demographic histories because iSMC infers the recombination landscape while jointly modelling ancestral coalescence rates. They could, however, be driven by ancestral recombination events that are shared between sampled individuals. This is because the number of such events is proportional to the relatedness between individuals and can lead to a signal of similarity in the inferred maps. Hence the recombination maps would be expected to be correlated in a manner that reflects the hominin phylogeny, even in the absence of evolution of the recombination landscape. In order to mitigate this effect, we took advantage of iSMC's explicit modelling of coalescence times to consider only recent recombination events. Because the discretisation of time is embedded in the hidden states of our HMM, we were able to restrict the posterior decoding to the first 10 time intervals (see Methods), thereby discarding recombination events older than ~0.25 coalescent time units and reducing the relevance of shared recombination events in our analysis. Overall, we were still able to recover the hominin phylogeny using pair-wise correlations between such time-restricted maps (**S3A Fig**), suggesting that the patterns depicted in **Fig 4** reflect the accumulation of differences in population-specific recombination landscapes over time. We further assessed whether the spurious signal of similarity imposed by shared recombination events would be strong enough to cause a reflection of the population

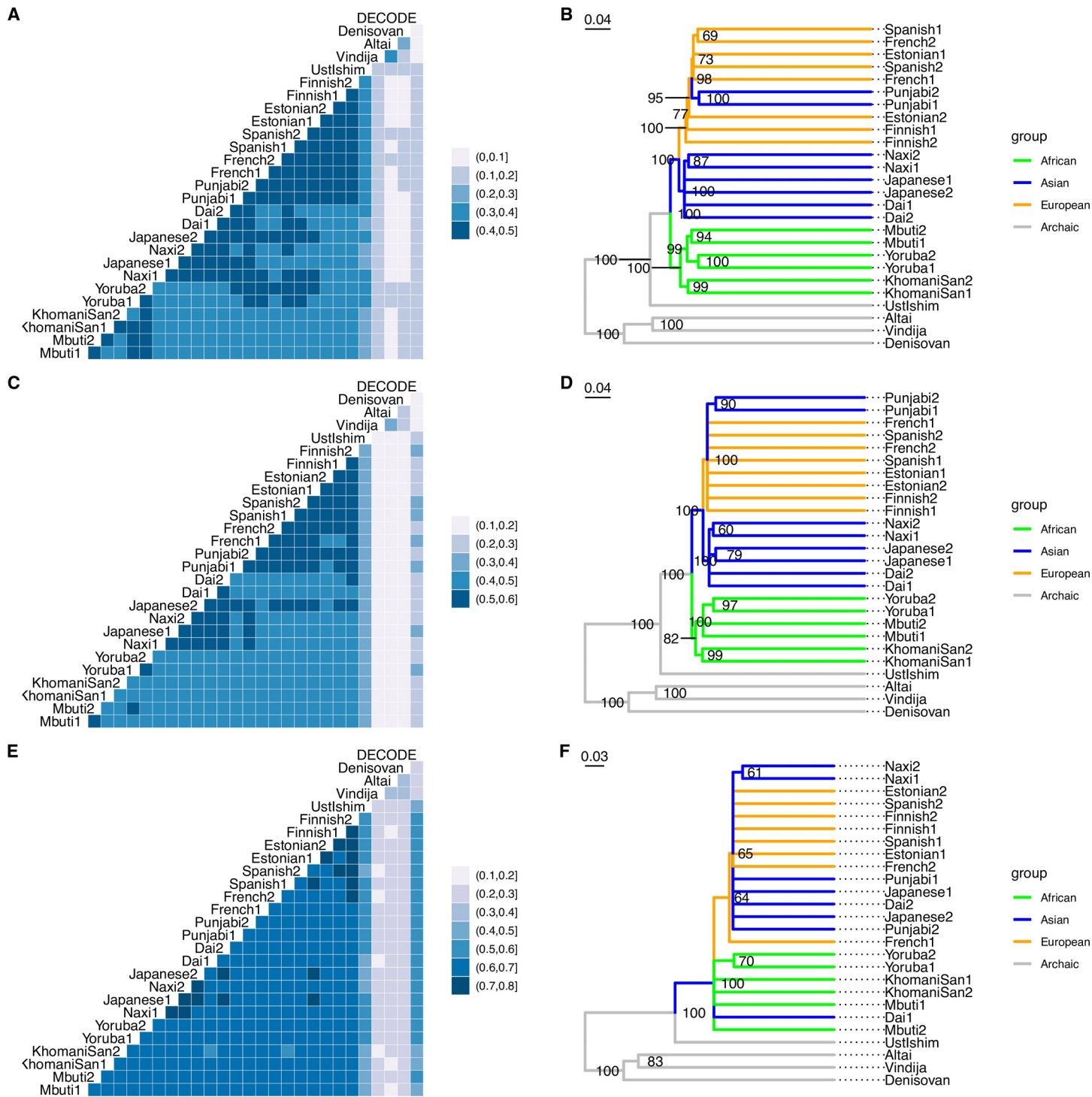

**Fig 4. Evolution of the recombination landscape in hominins.** Left: Correlograms depicting pair-wise Spearman's rank correlations between individual recombination maps. Right: corresponding dendrograms, highlighting the phylogenetic clustering of different groups (colour legend). The clades with less than 60% bootstrap support (over 1000 replicates) were collapsed. **A, B**: 50 kb windows. **C, D**:, 200 kb windows. **E, F**: 1 Mb windows.

phylogeny. To this end, we performed simulations following Fu et al.'s [61] study of a hominin-specific demographic history (see Methods). On top of the parameters chosen by Fu et al.,

we imposed a common recombination landscape to all simulated haplotypes and fitted iSMC to non-overlapping genomes pairs belonging to the same population. We then used the inferred maps at the 50 kb scale to build a dendrogram using the same procedure outlined above. Since the recombination landscape does not evolve in these simulations, then in the absence of a confounding signal, we would expect the similarity between such maps to be uniform across the populations resulting in a star phylogeny. Instead, we observe that the pair-wise correlations between such simulated maps still reflect the underlying phylogeny with relatively high bootstrap support for the major branches (S3B Fig), indicating the presence of substantial confounding. As a proof-of-principle that iSMC can disentangle these events, we obtained time-restricted recombination maps from the same simulated dataset. The correlation patterns of these maps no longer reflect the underlying history of populations, and the resulting dendrogram displays very low bootstrap support (S3C Fig). In summary, our results provide evidence that the divergence of the recombination landscape in hominins mirrors the divergence of the species.

## Discussion

Our improved SMC model is able to infer accurate recombination maps from high-quality pairs of genomes. Nevertheless, the agreement between iSMC and experimental maps in all three species studied is lower than that obtained in simulations. There can be several factors driving this difference. First, technical artefacts (e.g., sequencing errors and missing data) will increase the noise in the estimates, in particular at the fine scale. Second, introgression and natural selection [72] influence Ne locally but are unaccounted for by our model, thus introducing bias in the estimation of the compound parameter $\rho = 4 \times Ne \times r$. In particular, strong positive selection (or adaptive introgression) increases LD between loci, and the breadth of such effect should depend on the local recombination rate per nucleotide per generation, $r$. Such complex interplay can introduce severe biases in species where selection is widespread along the genome, particularly at narrow window sizes. Third, the distinct data types used by experimental and statistical methods imply that they measure different facets of recombination [13]. While experimental maps are a snapshot of the landscape at present-day generation, the historical map estimated by population genomic methods reflects the time-average cross-over rate at each position of the genome because ancient recombination events also influence the TMRCA distribution, as highlighted by our time-restricted analysis. As a result of this contrast between experimental and statistical approaches, the ensuing maps are not expected to be perfectly correlated, even in the absence of noise and model violations, because of the evolution of the recombination landscape (Fig 4). A promising perspective of future research using our model lies in the potential to focus the reconstruction of the recombination landscape to particular epochs, allowing more detailed investigation of its evolution.

Evolution of the recombination landscape implies that the present-day distribution of cross-over events may carry little information about linkage that influenced long-term processes such as linked selection and introgression. Therefore, historical maps are more meaningful than present-day maps in the context of assessing the evolutionary consequences of recombination rate variation [58]. Due to its power with restricted sample sizes, iSMC is well suited to extract LD information from population genomic datasets with high-quality whole-genome sequences from a small number of individuals [73–76]. We have demonstrated its accuracy in species with contrasting levels of diversity, demographic histories and selective pressures, and posit that it will be useful for investigation in other species. Not only will such maps aid the interpretation of diversity in non-model organisms, but a picture of the recombination landscape in different groups will tell us about the nature of recombination itself. Open questions include whether the recombination landscape is associated with large-scale genome

architecture and how variation in the recombination landscape relates to life history and ecological traits [77]. Finally, as ancient DNA samples become more common (including species other than humans [78]), it will be possible to obtain maps from extinct *taxa*, granting the opportunity to study the evolution of the recombination landscape with unprecedented resolution [44,79].

## Methods

### The Markov-modulated hidden Markov model framework

We now describe our current implementation of the model that focuses on the case of a single pair of genomes, noting that the Markov-modulated framework is general and can be extended to include multiple pairs. The original pair-wise SMC implementation [42] discretises a distribution of scaled coalescence times into $t$ TMRCA intervals to implement a discrete space Hidden Markov Model (HMM) with $t \times t$ transition matrix given by:

$$Q(\rho)_{smc} = \begin{bmatrix} G_{11} & G_{12} & \cdots & G_{1t} \\ G_{21} & G_{22} & & G_{2t} \\ \vdots & & & \vdots \\ G_{t1} & G_{t2} & \cdots & G_{tt} \end{bmatrix}$$ (1)

where $G_{ij}$ (the transition probabilities between genealogies $i$ and $j$, each being a TMRCA interval) is a function of ancestral coalescence rates and the global parameter $\rho$, which is assumed to be constant along the genome [39,42]. The key innovation we introduce in iSMC is to relieve this assumption by letting $\rho$ vary along the genome, following its own Markov process, where values drawn from a prior distribution are used to compute the transition probabilities between genealogies. (We use the equations of the SMC' model [39] as derived in [43].) Let $R$ be a strictly positive Gamma distribution with a single parameter ($\alpha = \beta$, which constrains the distribution to have mean equal to 1.0) describing the variation in recombination along the genome. The shape of this prior distribution is flexible since $\alpha$ is estimated by iSMC together with demographic parameters. If $R$ is discretised into $k$ categories of equal density, the possible values that $\rho$ can assume in the Markov-modulated process are all $r_j \times \rho_0$, where $r_j$ is the mean value inside the $j$th $R$ category and $\rho_0$ is the genome-wide average population recombination rate. Our Markov model (inspired by the observation that the distribution of cross-over events is not random, but clustered in regions of similar values) states that the probability distribution of $R$ at position $i + 1$ only depends on the distribution at position $i$. We consider the case where the transition probability between any two $R$ categories ($P_{ij}$) is identical and equivalent to one auto-correlation parameter ($\delta$), which is also estimated by iSMC. The transition matrix of this Markovian process is simply:

$$Q_\rho = \begin{bmatrix} P_{11} & P_{12} & \cdots & P_{1k} \\ P_{21} & P_{22} & & P_{2k} \\ \vdots & & & \vdots \\ P_{k1} & P_{k2} & \cdots & P_{kk} \end{bmatrix} = \begin{bmatrix} 1-\delta & \dfrac{\delta}{k-1} & \cdots & \dfrac{\delta}{k-1} \\ \dfrac{\delta}{k-1} & 1-\delta & & \dfrac{\delta}{k-1} \\ \vdots & & & \vdots \\ \dfrac{\delta}{k-1} & \dfrac{\delta}{k-1} & \cdots & 1-\delta \end{bmatrix}$$ (2)

Because ρ is a parameter of the SMC, spatial variation in the recombination rate affects the transition probabilities between genealogies ($\mathbf{Q_{SMC}}$). Since spatial variation in ρ is itself modelled as a Markovian process, the combined process is said to be Markov-modulated by $\mathbf{Q_{\rho}}$, leading to a Markov-modulated HMM. If $t$ is the number of discretised TMRCA intervals of the SMC, and $k$ is the number of discretised ρ categories, then the Markov-modulated HMM is an HMM with $n = t \times k$ hidden states (**Fig 1**). The transition matrix of the Markov-modulated process, $\mathbf{Q_{iSMC}}$, has dimension $n \times n$ and is obtained from $\mathbf{Q_{\rho}}$ and $\mathbf{Q_{SMC}}$:

$$Q_{iSMC} = \begin{bmatrix} P_{11} \cdot Q(\rho_{j=1})_{SMC} & \cdots & P_{1k} \cdot Q(\rho_1)_{SMC} \\ \vdots & & \vdots \\ P_{k1} \cdot Q(\rho_{j=k})_{SMC} & \cdots & P_{kk} \cdot Q(\rho_k)_{SMC} \end{bmatrix} \tag{3}$$

In brief, $\mathbf{Q_{iSMC}}$ is a composition of $k^2$ sub-matrices of dimension $t \times t$, each being a $\mathbf{Q_{SMC}}$ assembled using $\rho_0$ scaled the corresponding category of $R$ to compute transition probabilities between genealogies. The main diagonal sub-matrices are further scaled by $1 - \delta$, and the off-diagonal sub-matrices by $\delta / (k - 1)$. The Markov structure allows efficient integration over all recombination rates by means of the forward recursion (**Eq 4**).

### Model selection and computation of the posterior recombination landscape

iSMC works in two steps: (1) fitting models of recombination rate variation and demography, and (2) inferring recombination maps based on the selected model. During step 1, the model parameters (ρ, δ, α and a set of splines controlling the shape of the demography, see below) are optimised by maximizing the likelihood using the Powell multi-dimensions procedure [80], which is computed for the entire sequence by applying the forward recursion of the HMM [48] as implemented in the zipHMMlib [81] at every position $i$ of the alignment:

$$F_{i,G_v}(\rho_m) = (\sum_{l=1}^{k}(\sum_{u=1}^{t} F_{i-1,G_u}(\rho_l) \times (G_{uv}|\rho_l) \times \rho_{Im})) \times (G_v \rightarrow S_i) \tag{4}$$

where we integrate over all $k$ discretised values of ρ and over all $t$ TMRCA intervals. The transition between genealogies ($G_{uv}$) is a function of both the focal recombination rate ($\rho_l$) and the ancestral coalescence rates, whereas $G_v \rightarrow S_i$ represents the emission probability from $G_v$ to the observed state at position $i$–which is greatly simplified in the case of a single pairs of genomes [42]. In case AIC favours the heterogeneous model, in step 2 iSMC uses the estimated parameters to calculate the posterior average ρ for all sites in the genome. To this end, it first uses the so-called posterior decoding method [48] as implemented in zipHMMlib [81] to compute the posterior probability of every hidden state at each position in the sequence. Since in the Markov-modulated HMM the hidden states are pairs of ρ categories and TMRCA intervals (**Fig 1C**), this results, for all sites $i$ in the genome, in an $n$-dimensional vector with posterior probabilities for each hidden state. We can interpret these vectors as joint probability distributions $P_i(x,y)$ where $\{x_l\}_{1 \leq l \leq k}$ is the $k$-dimensional vector of recombination rates (**Fig 1B**) and $\{y_j\}_{1 \leq j \leq t}$ is the $t$-dimensional vector of TMRCA intervals (**Fig 1A**). Thus, if $r_l$ is the value of $R$ inside discretised category $l$ and $\rho_0$ the genome-wide average recombination rate, the posterior average ρ at position $i$ is given by

$$\bar{\rho}_i = \rho_0 \cdot \sum_{l=1}^{k} r_l \cdot \sum_{j=1}^{t} P_i(x_l, y_j) \tag{5}$$

## Time-restricted posterior decoding

In the computation of $\rho_i$ described in Eq 5, we obtain the marginal probability distribution of recombination rates by summing the posterior probabilities of each category $l$ over all TMRCA intervals. To focus the inferred maps on the recent past, we further constrain the computation of $\rho_i$ using a three-step procedure: first, we decode site-specific TMRCA's by choosing the interval with maximum posterior probability, integrating over all recombination rate classes; second we compute site-specific posterior-average recombination rates by considering the probability of each category of the Gamma prior exclusively inside the most likely TMRCA. The posterior average $\rho$ for site $i$ is then given by

$$\bar{\rho}_i = \rho_0 \cdot \sum_{l=1}^{k} r_l \cdot P_i(r_l | \tau = x) \tag{6}$$

where $x$ is the most likely TMRCA at site $i$. These first two steps lead to a reconstructed landscape consisting of a TMRCA and the average recombination rate within such TMRCA, for each site in the genome. Finally, when binning recombination maps, for each window we only compute the average $\rho$ over sites with a TMRCA within a predefined range, e.g., $0 < \tau < 10$.

## Testing hidden state configurations

The hidden states of iSMC are pairs of genealogies and recombination rates where both elements are drawn from discretised distributions. Since the complexity of the forward algorithm (which computes the likelihood) is quadratic in the number of hidden states, there is a limit to the discretisation scheme that can be adopted, as too fine a discretisation would lead to impractical execution times. We thus fixed the number of hidden states to 200 and used this limit to run iSMC in all simulated datasets. Within this maximum, however, it is possible to devise several combinations of hidden states by changing the way in which we discretise the TMRCA and $\rho$ distributions. The goal of reconstructing the recombination landscape would in principle make natural the choice of investing in a fine-grained discretisation of the distribution of $\rho$. However, this would mean a coarse-grained discretisation of time and, since the signal for fitting the distribution of $\rho$ comes from the expected number of TMRCA transitions, this strategy could reduce iSMC's power to detect such changes. Therefore, in the simulation study, we tested the performance of two configurations: 20 TMRCA intervals and 10 $\rho$ categories and 40 TMRCA intervals and 5 $\rho$ categories (S2 Table).

## Modelling complex demographic histories

HMM implementations of the SMC [41–43] typically use the expectation-maximisation algorithm to optimise transition probabilities, where the actual targets of inference–the coalescence rates at each time interval–are latent variables of the model. Here we use cubic spline interpolation (similarly to [41]) to map coalescence rates at time boundaries, which are then assumed to be piecewise constant for the duration of each interval. Because we use three internal splines knots (i.e., the demographic history is divided into four epochs wherein a cubic curve is fitted), the number of parameters is substantially reduced in our model–in particular when a fine discretisation of TMRCA is employed. Importantly, in the spline implementation, the number of model parameters is independent of the number of classes in the discretisation scheme.

## Simulation study

**Four scenarios of spatial variation in $\rho$.** We simulated a piece-wise constant recombination rate along the genome by drawing values from a continuous Gamma distribution with parameters $\alpha$ and $\beta$, and segment lengths from a geometric distribution with mean length g.

We considered four possible scenarios where $\alpha = \beta = 0.5$ or $5.0$, and $g = 100$ kb or 1 Mb. For each of the four combinations, we simulated 10 independent pairs of two 30 Mb haploid chromosomes under a constant population size model, assuming $\theta = 0.003$ and $\rho_0 = 0.0012$. For each of the following simulated scenarios, we focus on the landscape with $\alpha = 0.5$ and $g = 100$ kb. All scenarios share the same sequence length, sample size, as well as $\theta$ and $\rho$ parameter values.

**Demographic history.** We simulated two demographic scenarios. First, a 20-fold population expansion 0.01 coalescent time units ago; second, a 20-fold population bottleneck 0.5 coalescent time units ago. Assuming an effective population size of 30,000, these coalescent times correspond to 1,200 and 60,000 generations ago, for the expansion and bottleneck events, respectively.

**Introgression events.** We simulated two introgression scenarios where a source population introduces a pulse of genetic material into a target population. In both scenarios, the split between source and target populations happened 2.0 coalescent time units ago, and the source replaces 10% of the genetic pool of the target. In the first scenario, secondary contact happened 0.125 coalescent time units ago; in the second, it happened 0.25 coalescent time units ago. Assuming an effective population size of 30,000, the split between population happened 240,000 generations ago, and the introgression events happened 15,000 and 30,000 generations ago, respectively.

**Variation in the mutation rate.** We simulated a piece-wise constant mutation rate along the genome by drawing rate values from a uniform distribution between 0.1 and 10.0 and segment lengths from a geometric distribution with mean length f, where f is either 20 kb or 500 kb. The uniform distribution generating scaling factors of $\theta$ has a mean equal to 5.05 instead of 1.0. In this case, the expected genome-wide average $\theta = 0.015$. The reason for that is our focus on the spatial distribution of $\theta$ itself. If the landscape had mean = 0.003, its highly heterogeneous nature would scale $\theta$ down to values well below $\rho$ (0.0012) too often along the 30 Mb sequence. The ensuing loss of signal (due to low SNP density) would result in poorly inferred maps that display low correlations with the simulations, not because of spatial *heterogeneity* in $\theta$ (local transitions), but instead because the ratio $\theta / \rho$ would be too low in many windows across the chromosome. In all the above scenarios, the proportion of variance ($R^2$) in simulated maps that is explained by inferred maps was computed after binning the landscapes into non-overlapping windows of 50 kb, 200 kb, 500 kb and 1 Mb, that is, the analysis is agnostic to the true breakpoints of the simulated landscapes.

**Hominin demography.** We simulated 100 Mb-long haploid chromosomes according to sample size and parameters provided by Fu et al. [61]. We chose this sequence length because focusing on a 100 Mb segment of chromosome 1 was sufficient to reconstruct the hominin phylogeny based on the recombination maps. In other words, there is enough signal on a 100 Mb segment of human genomes. We added a simulated recombination landscape with $\alpha = \beta = 0.5$ and $g = 100$ kb to the ms arguments provided by the authors.

## Data analyses

Model selection followed by inference of recombination maps in *D. melanogaster* and *Z. tritici* was performed using publicly available data (chromosome 2L from haploid pairs ZI161 / ZI170, ZI179 / ZI191 and ZI129 / ZI138 in the Drosophila Population Genomics Project Phase 3 [82]; chromosome 1 from haploid pairs Zt09 / Zt150, Zt154 / Zt155 and Zt05 / Zt07 for *Z. tritici* [64]. Gaps and unknown nucleotides in these FASTA sequences were assigned as missing data. $R^2$ were computed as the square of the Pearson correlation coefficient between the resulting (binned) recombination landscape and available genetic maps, and their 95% confidence

intervals (CI) estimated using 100 bootstrap replicates by re-sampling map windows with replacement. For the benchmarking in *Z. tritici*, we used the average between the two crossover maps given in [62].

Individual IDs for the 22 modern human samples from the Simons Genome Diversity Panel [66] are presented in **S6 Table**. Data were downloaded from Harvard's public repository (https://sharehost.hms.harvard.edu/genetics/reich_lab/sgdp/vcf_variants/vcfs.variants.public_samples.279samples.tar) in July 2018. We used the available strict mask to set low-quality regions as missing data. The four ancient DNA samples were downloaded from the server at the Max Planck Institute for Evolutionary Anthropology in Leipzig (http://cdna.eva.mpg.de/neandertal/Vindija/VCF) in May 2018. Details of the model fitting procedure in the ancient samples are described in the Results section. Based on estimates obtained from extant humans (**S1 Fig**), posterior decoding was performed by fixing ρ to 0.00015, α to 0.5 and δ to 0.000025. After binning the maps, all downstream analyses were conducted after discarding windows with more than 50% missing data in any of the hominin sequences and considering only positions present in the DECODE genetic map. The correlograms and dendrograms presented in **Fig 4** were obtained by hierarchical clustering (using UPGMA) of pair-wise distances computed from $1 - r_s$, where $r_s$ is the Spearman correlation of ranks between two individual recombination maps. The clades with less than 60% of bootstrap support (over 1000 replicates) were collapsed in the dendrograms. Recombination maps for all samples are available as a resource on FigShare under the DOI: http://10.0.23.196/m9.figshare.8269703.

## Computing performance

The limiting computing resources are different between the optimisation and decoding steps performed by iSMC. During optimisation, execution time is key: the program typically takes about two weeks to fit a 40x10 model on the largest human chromosome, using a standard desktop machine. iSMC treats chromosomes independently, meaning that if a whole-genome sequence is input, it can parallelise computation across the multiple chromosomes. Therefore, fitting the model on the whole-genome sequence of a human individual takes roughly the same time as it takes to fit the model on its largest chromosome, provided that enough CPUs are available for full parallelisation. During computation of the posterior average recombination rate, the limiting resource is memory. On the same desktop machine and under the same 40x10 model, it takes around 20 min and 24 Gb of RAM to decode a 2 Mb fragment of the genome. Since these operations can also be parallelised over distinct chromosomes, decoding the full human genome uses a peak 528 Gb of RAM (24 Gb for each of 22 chromosomes) and takes about 42 hours to complete (20 min for each of the 125 windows of size 2 Mb in its largest chromosome). Crucially, the optimisation and decoding steps can be decoupled, such that the user can estimate parameters on a first run of the program and (arbitrarily) later decode the recombination maps. This separation between optimisation and posterior decoding allows the user to request the appropriate computer resources at each step of the inference, e.g., when using a computing cluster.

We note that both the speed and accuracy of the optimisation depend on the overall genetic diversity as well as on the proportion of "missing data" in the sequences. On the one hand, higher density of SNPs should increase iSMC's ability to identify recombination events, thus traversing the likelihood surface more efficiently and taking less optimisation steps to find a local optimum. On the other hand, large amounts of missing data will "flatten" the likelihood surface, therefore reducing the power to distinguish between likelihood peaks and taking more time to converge. We recommend running iSMC on datasets with less than 15% of missing data.

## Supporting information

**S1 Table. AIC value for each replicate of each of the simulated landscapes.**
(ODS)

**S2 Table. $R^2$ for each replicate of each parameter of the simulated landscape, according to discretisation scheme.**
(ODS)

**S3 Table. $R^2$ for each replicate of each simulated demographic history, according to whether coalescence rates were jointly-inferred or not.**
(ODS)

**S4 Table. $R^2$ for each replicate of each scenario of introgression.**
(ODS)

**S5 Table. $R^2$ for each replicate of each scenario of mutation rate variation.**
(ODS)

**S6 Table. Metadata of the extant human samples.**
(ODS)

**S1 Fig. Estimates of recombination-related parameters in modern human samples. A:** the genome-wide recombination rate ($\rho$); **B:** the shape of the Gamma distribution ($\alpha$); **C:** the average frequency of change in the recombination rate along the genome ($\delta$).
(PDF)

**S2 Fig. $R^2$ for each replicate as a function of the $\theta$ / $\rho$ ratio.**
(PDF)

**S3 Fig. Dendrograms computed from pair-wise similarities in recombination maps at the 50 kb scale. A:** time-restricted maps based on human whole-genome sequences; **B:** regular (non time-restricted) maps based on a simulation of hominin specific demography; **C:** time-restricted maps based on the same data as **B**.
(PDF)

**S1 File. R scripts required to reproduce our analyses.**
(GZ)

## Acknowledgments

The authors thank Alice Feurtey, Asger Hobolth, Bernhard Haubold, Eric Hugoson, Eva Stukenbrock, Fabian Klötzl, Kai Zeng, Pier Palamara and Stephan Schiffels for fruitful discussions about this work.

## Author Contributions

**Conceptualization:** Gustavo V. Barroso, Julien Y. Dutheil.

**Formal analysis:** Gustavo V. Barroso, Nataša Puzović, Julien Y. Dutheil.

**Funding acquisition:** Julien Y. Dutheil.

**Investigation:** Gustavo V. Barroso, Julien Y. Dutheil.

**Methodology:** Gustavo V. Barroso, Julien Y. Dutheil.

**Project administration:** Julien Y. Dutheil.

**Software:** Gustavo V. Barroso.

**Supervision:** Julien Y. Dutheil.

**Writing – original draft:** Gustavo V. Barroso, Julien Y. Dutheil.

**Writing – review & editing:** Gustavo V. Barroso, Julien Y. Dutheil.

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
