## [Decision Letter · Decision Letter 0]

19 Jul 2019

Dear Dr Valadares Barroso,

Thank you very much for submitting your Research Article entitled 'Inference of recombination maps from a single pair of genomes and its application to ancient samples' to PLOS Genetics. Your manuscript was fully evaluated at the editorial level and by independent peer reviewers.

As you will see, the reviewers were all quite positive in their assessments, but there are several issues that should be addressed. We therefore ask you to modify the manuscript according to the review recommendations before we can consider your manuscript for acceptance. Your revisions should respond to the specific points made by each reviewer. A carefully revised manuscript might not need to be seen again by the reviewers. I will make that judgment based on the quality of your revisions.

[LINK]

Yours sincerely,

Bret Payseur

Section Editor: Evolution

PLOS Genetics

Reviewer's Responses to Questions

**Comments to the Authors:**

Reviewer #1: In this paper Barroso and colleagues present a novel method, based on the sequentially Markov coalescent, that can jointly infer spatial variation in recombination rate and past population sizes. The paper represents an advance in both the area of coalescent hidden Markov models--it is the first model to attempt to account for spatial heterogeneity in this framework--and recombination rate inference--making inference of recombination rates possible from even single diploid samples. Both of these contributions are important and will be useful to a broader audience of population geneticists, especially empiricists working in non-model organisms and the ancient DNA community. The paper is well-written, enjoyable to read, and represents and important advance but I do have a small number of comments that I list below.

Major Comments:

-On p. 13 the authors claim "these results cannot be accounted for by patterns of LD left by population-specific demographic histories because iSMC infers the recombination landscape while jointly modeling ancestral coalescence rates". I think that this statement conflates two issues, and is misleading. On the one hand, bias introduced by assuming a constant demography (see [1,2,3]; references listed at the end of this review) could cause unrelated populations with similar past population size histories to have similar inferred recombination rates--by accounting for past population sizes iSMC certainly deals with this potential issue. On the other hand, this issue is distinct from the more subtle genealogical LD hinted at in the manuscript. There is some "true" ARG relating all organisms (see [4]), and recombinations that happen in ancient parts of this ARG are likely shared by many extant individuals. The more closely related individuals are, the more likely they are to share these ancient recombination events. This would cause the inferred recombination rates to be correlated, even if the true underlying recombination rates are not (see the discussion in [5]). No method that infers historical recombination rates, including iSMC, can avoid this confounding. And particularly, because iSMC is using the signal from only a pair of haplotypes and so most events are expected to occur on the order of 1 coalescent unit ago (i.e. fairly anciently) many of these events will be shared across different samples, meaning that this confounding should be particularly prevalent in iSMC.

-In a previous version of this manuscript (https://www.biorxiv.org/content/10.1101/452268v2) the authors included a discussion of the runtime, which is absent in the present manuscript. Any paper presenting a computational method should provide an empirical analysis of the runtime of the method (e.g., how long did it take to perform the whole genome analyses for the humans in the present manuscript?). In general, a runtime of days or weeks is not prohibitive, but users should certainly be aware of what to expect from reading the manuscript. There are also a number of potential speedups to the HMM that while certainly not necessary for publication, could dramatically improve the runtime of iSMC, which would make the method more broadly useful. Specifically, the transition matrix Q_rho has a special structure where conditioned on changing recombination rates, it is equally likely to change to any other recombination rate. In a number of previous methods [6, 7, 8], the exact same structure (although in a different context) has been exploited by computing the transition as a combination of the transition conditioned on changing state (whereby it does not matter which state the HMM was in previously) and conditioned on not changing state (whereby the state can only remain in the same state). Such considerations should reduce the runtime from quadratic to linear in the number of recombination rate bins. Similarly, the genealogical part of the HMM has sufficient structure to reduce the runtime from quadratic to linear (e.g., [9, 10]). An alternative approach is to use the spectral decomposition method of [11], which allows for the "skipping'" of homozygous sites.

-On pp. 13-14 there is a discussion of why the correlation is higher in simulations than in the real data. While natural selection, introgression, and evolution of the recombination rate certainly play a role, there are simpler reasons for the improved performance on simulated data (discussed below in the minor comments). In particular, the evidence in favor of evolving recombination rates is presented in terms of the patterns of correlation at different scales in the real data and in simulated data. This is not a fair comparison because the r^2 on the simulated data is likely inflated for reasons discussed below and since r^2 is bounded by 1, the slope must necessarily be less steep.

Minor Comments:

-p. 8 and pp. 10-11, results should be reported for each window size (50kb, 200kb, 500kb, 1Mb), instead of just the median across window sizes on p. 8, and for a single window size for pp. 10-11. Furthermore, since hotspots are on the order of kb in length it would be good to compute these metrics at even finer scales such as 1kb.

-p. 10 Saying that the "variance explained is significant" should be removed. Proportion of variance explained is non-negative, and so it only takes on the value 0 if the observed correlation (r) is exactly 0. For continuous observations, this should happen on a set of measure 0, and so no bootstrap-based test would fail to reject this null. A better test would be to see if the bootstrap confidence interval for the correlation itself (i.e., r instead of r^2) contains 0. Furthermore, by providing the observed proportions of variance explained (with confidence intervals) the authors already convey this information.

-The presented simulation results are biologically unrealistic in such a way that would optimize the performance of iSMC. In particular, in humans and many other species the ratio of the mutation rate to the recombination rate is approximately 1, whereas in the simulations this ratio is >2, which will facilitate inference. This is further exacerbated in the variable mutation rate case where the average mutation rate over the recombination rate is further increased to >10. The length scale of changing recombination rates is also rather unrealistic, with changes occurring either every 100kb or 1Mb on average. Because hotspots tend to be on the order of a few kb in species with PRDM9, the length scales used in simulation are 2-3 orders of magnitude too large to accurately reflect realistic recombination maps. These smoother true recombination maps would make inference substantially easier. Additionally, the simulations involving introgression only test fairly ancient introgression -- for human-like values of Ne and generation time the more recent introgression simulation would correspond to an introgression time of about 70kya. A far more concerning issue in practice would be recent gene flow which would affect longer chunks of the genome, resulting in the sort of long-range LD patterns discussed in the manuscript. It is not absolutely necessary to perform new simulations to account for any of these issues (elevated mutation rate to recombination rate ratio, overly smooth true recombination maps, and ancient introgression events) but it would be good to explicitly point out that these simulation scenarios suggest a much better performance of iSMC than is likely to be seen in practice (see the aforementioned major comment).

-The performance of the method on data from regions under selection is never benchmarked. While weak background selection can be approximated by a local reduction in Ne in terms of the amount of polymorphism, it is unclear that this should have a monotonic, linear effect on the inferred rho as implied on p. 13. The effect is likely to be even more complex in the presence of recent selective sweeps, stronger background selection, etc... I do not think that this is a major issue or needs to be directly addressed (e.g. by a simulation study), but perhaps a note of caution should be included, especially for species where selection is likely to be pervasive.

Typos and very minor comments:

-p. 3 "where a survey of the recombination landscape in closely related species are lacking"  "where a survey of the recombination landscape in closely related species is lacking"

-p. 8 "and the short length of blocks with constant [rho] leaves few signal in the data"  "and the short length of blocks with constant [rho] leaves little signal in the data"

-p. 12, the authors discuss dealing with the ancient samples in a different way by first inferring a demography and mean recombination rate, and then using parameters inferred from modern humans as a prior for the posterior decoding. This approach seems reasonable, but it should bias the inferred maps to be closer to modern humans. Since the ancient individuals still form a clade to the exclusion of modern humans, such bias would only imply that Neanderthals and Denisovans have even more diverged recombination maps. It would be good to discuss that this bias is in the "conservative" direction and so it does not affect the major conclusions of this section.

References:

[1] H. R. Johnston and D. J. Culter, "Population demographic history can cause the appearance of recombination hotspots". The American Journal of Human Genetics, 2012.

[2] J. A. Kamm, J. P. Spence, J. Chan, Y. S. Song, "Two-locus likelihoods under variable population size and fine-scale recombination rate estimation". Genetics, 2016.

[3] A. L. Dapper, B. A. Payseur, "Effects of demographic history on the detection of recombination hotspots from linkage disequilibrium". Molecular Biology and Evolution, 2017.

[4] P. L. Ralph, "An empirical approach to demographic inference with genomic data". Theoretical Population Biology, 2019.

[5] J. P. Spence and Y. S. Song, "Inference and analysis of population-specific fine-scale recombination maps across 26 diverse human populations". BioRxiv, 2019.

[6] P. Fearnhead and P. Donnelly, "Estimating recombination rates from population genetic data". Genetics, 2001.

[7] N. Li and M. Stephens, "Modeling linkage disequilibrium and identifying recombination hotspots using single-nucleotide polymorphism data". Genetics, 2003.

[8] S. Sheehan, K. Harris, and Y. S. Song, "Estimating variable effective population sizes from multiple genomes: a sequentially Markov conditional sampling distribution approach". Genetics, 2013.

[9] K. Harris, S. Sheehan, J. A. Kamm, and Y. S. Song, "Decoding coalescent hidden Markov models in linear time". RECOMB, 2014.

[10] P. F. Palamara, J. Terhorst, Y. S. Song, and A. L. Price, "High-throughput inference of pairwise coalescence times identifies signals of selection and enriched disease heritability". Nature Genetics, 2018.

[11] J. Terhorst, J. A. Kamm, and Y. S. Song, "Robust and scalable inference of population history from hundreds of unphased whole genomes". Nature Genetics, 2017.

Reviewer #2: This paper presents a useful new method for jointly inferring population size changes with the recombination rate landscape of a single genome. The method is very well described, and based on benchmarking studies summarized by the authors, it infers maps that are at least as well correlated with pedigree-based maps as those produced by state of the art methods that require input data from many genomes. I do think that although the results presented on pages 10 and 11 are impressive, they don’t seem strong enough to justify the section heading “iSMC outpeforms the state of the art…” Although the point estimate of variance explained by the iSMC is higher than the point estimate of variance explained by LD helmet, the confidence interval of the iSMC estimate includes the LD helmet estimate. Is there a stronger case to be made that the iSMC is doing better than LD helmet overall? Or do certain genomes yield better estimates because those genomes are closely related to the genomes that were used to make the family-based estimates?

One parameter of the method that is not very well described is the gamma prior probability distribution that the recombination rate will change from one locus to the next. There’s nothing in the coalescent model that dictates how fast the recombination rate is likely to change as a function of sequence, so this choice seems somewhat arbitrary and might not be well optimized as is. Unless I missed it, the manuscript doesn’t seem to discuss how the shape and scale of the gamma distribution have been chosen, and it seems worth exploring whether varying them could improve iSMC’s performance. Figure 3E-H suggests that the inferred recombination landscape is systematically too spikey and biased toward high recombination rates, and perhaps this problem could be fixed or reduced by lowering the prior probability of recombination rate change between loci.

It is interesting to see that recombination landscapes cluster within human populations, but I’m concerned that this result could be misleading because of misidentification of genetic drift or demographic change as recombination landscape change. This result would be more convincing if it were accompanied by a proof of concept simulation study. If you were to simulate data from human populations that diverged under a realistic demographic scenario but share the same recombination map, would iSMC infer similar recombination maps from all individuals or instead infer some variation that comes from misidentified demographic variation? In other words, would inferred recombination maps from these populations also appear to cluster by population even though this structure is not built into the simulation?

Reviewer #3: This paper presents a method, iSMC, that is adapted from the PSMC framework and allows for the joint inference of demography and spatial variation in recombination rate along the genome. The method applies to a single unphased diploid sample, and aims to uncover a recombination map in order to aid studies of recombination in non-model systems. The paper considers an interesting problem, and while there are important caveats, the results are generally promising, particularly it seems for species with high diversity.

Beginning with the positive results, the authors demonstrate that iSMC has fairly good to very good squared correlations between their inferred maps and the true map in simulated data. The real data results from Z. tritici are not great, but those from D. melanogaster are reasonably good and the correlations of both are better than those from LDhat given data from more than one sample.

The results from humans are less reliable, with R^2 to the deCODE map of 0.166 at the 50 kb scale and 0.550 at the 1 Mb scale. While the text says that, “This suggests that the differences in R^2 between simulation and case studies are not only driven by noise, but also by biological factors,” this is hard to be certain of. For example, the simulations use higher diversity rates with a genome-wide average theta of 0.015 (pg. 21), or >10-fold higher than human diversity. Since SNPs are the fundamental signal for the analysis, the simulation results need not be informative of the results in humans. Moreover, the deCODE, pedigree-based map is strongly correlated with the LD-based HapMap map (Kong et al. 2010) -- with a correlation (seemingly just R) of 0.729 at the 10 kb scale and 0.920 at 3 Mb.

Given the above, the focus of the manuscript on ancient hominin samples is hard to justify. Instead, the authors could focus on species and potentially ancient samples that have a much higher diversity rate. A related concern is the inferred phylogenies, particularly the locations of the Ust’Ishim ancient sample, which seems more reasonable to place as an out-of-Africa sample, but instead is a sister group to other modern humans.

To help understand the contexts in which iSMC can function, providing more detailed simulation results including a wider range of theta (diversity) values, and potentially expanding on the demographic scenarios tested would be informative.

Lastly, please quote runtimes of iSMC.

Minor:

Pg. 2: “sequences from the ancestral population” - here, “ancestral population” is not well defined and could be clarified.

Pg. 5: “extending the SMC to accommodate spatial heterogeneity in rho ...” - this is about PSMC, not the SMC, right?

Pg. 6: “potentially leading to more accurate estimates” - it seems like the text here could be a bit more specific and conclusive about the estimation accuracy. More specifically, the word “potentially” could be omitted and the context of greater accuracy provided.

Pgs. 8-10: while this is in the Results section, a few more details about the simulation scenarios, such as the meaning of the A (alpha) and g parameters (Fig 2) and what the simulation parameters are would be good to include here. (Also see pg. 12, which mentions rho, alpha, and delta without definition.)

Pg. 12 “with 100% bootstrap support (with one exception, see below)” - please give the true % including the exception.

Eq. (3): the rho_i notation doesn’t seem to be introduced; only rho_0 and r_j.

Eq. (4): the probabilities here, e.g., Pr(rho_l -> rho_m), use a different notation than earlier in the text, where P_lm represents this probability. Would be helpful to be more consistent.

Fig. 3: While it seems likely that A-D and E-H represent increasing bin sizes, it would be good to make this more explicit.

Pg. 11 / citation 67: This is not the high coverage Vindija Neanderthal paper. See https://science.sciencemag.org/content/358/6363/655.

**Have all data underlying the figures and results presented in the manuscript been provided?**

Reviewer #1: Yes

Reviewer #2: Yes

Reviewer #3: Yes

PLOS authors have the option to publish the peer review history of their article (what does this mean?). If published, this will include your full peer review and any attached files.

Reviewer #1: Yes: Jeffrey P. Spence

Reviewer #2: No

Reviewer #3: No

---

## [Decision Letter · Decision Letter 1]

30 Sep 2019

Dear Dr Valadares Barroso,

We are pleased to inform you that your manuscript entitled "Inference of recombination maps from a single pair of genomes and its application to ancient samples" has been editorially accepted for publication in PLOS Genetics. Congratulations!

Please note the comment from Reviewer 3.

Yours sincerely,

Bret Payseur

Section Editor: Evolution

PLOS Genetics

Comments from the reviewers (if applicable):

Reviewer's Responses to Questions

**Comments to the Authors:**

Reviewer #1: The authors have addressed all of my concerns.

Reviewer #2: The authors have revised the article to my satisfaction. I have no further comments.

Reviewer #3: The authors have addressed my prior concerns, including more thorough evaluations of iSMC’s performance for smaller values of theta / rho. The model and paper are both very interesting. I have only one comment (and two other textual comments).

On pg. 116-9: “In the SMC, transition probabilities between genealogies are functions of ancestral coalescence rates and – of key relevance to this study – the population recombination rate (ρ) [42,43]. An important limitation of current implementations, however, is that they neglect variation in ρ along the genome and model its genome-wide average instead.”

The above isn’t true of ARGweaver, which uses an input genetic map to aid inference. This is worth noting.

Minor:

Line 133: “Coalescent” need not be capitalized.

Line 348-9: extra “of” in “to non-overlapping of genomes pairs belonging to the same population”

**Have all data underlying the figures and results presented in the manuscript been provided?**

Reviewer #1: Yes

Reviewer #2: Yes

Reviewer #3: Yes

PLOS authors have the option to publish the peer review history of their article (what does this mean?). If published, this will include your full peer review and any attached files.

Reviewer #1: Yes: Jeffrey P. Spence

Reviewer #2: No

Reviewer #3: No

**Data Deposition**

http://datadryad.org/submit?journalID=pgenetics&manu=PGENETICS-D-19-00987R1

**Press Queries**

---

## [Editor Report · Acceptance letter]

22 Oct 2019

PGENETICS-D-19-00987R1 

Inference of recombination maps from a single pair of genomes and its application to ancient samples 

Dear Dr Valadares Barroso, 

We are pleased to inform you that your manuscript entitled "Inference of recombination maps from a single pair of genomes and its application to ancient samples" has been formally accepted for publication in PLOS Genetics! Your manuscript is now with our production department and you will be notified of the publication date in due course.

With kind regards,

Matt Lyles

PLOS Genetics

On behalf of:
